# The Extreme Ends of the Treatment Response Spectrum to Botulinum Toxin in Cervical Dystonia

**DOI:** 10.3390/toxins13010022

**Published:** 2020-12-31

**Authors:** Sara Samadzadeh, Raphaela Brauns, Harald Hefter

**Affiliations:** Department of Neurology, University Hospital of Düsseldorf, Moorenstrasse 5, D-40225 Düsseldorf, Germany; sara.samadzadeh@yahoo.com (S.S.); montanabrauns@t-online.de (R.B.)

**Keywords:** botulinum neurotoxin, response spectrum, hypersensitivity, insensitivity, response behavior, neutralizing antibody

## Abstract

Background: The response to BoNT is not uniform; a broad spectrum of responses and side-effects usually occurs. This study aimed to show special cervical dystonia cases with therapy response very different to normal treatment course which indicate the extreme ends of therapy spectrum. Patients: Clinical data and course of treatment of five long-term treated patients with cervical dystonia out of therapy response norms are presented: a patient who was supersensitive to standard dose and has required dose adjustment to lower dose of BoNT; one patient who worsened under a standard dose, but responded excellently to twice the standard dose; one insensitive patient who responded poorly for years to a dose well above the standard dose, but responded when dose was further increased; and two patients with a totally different response pattern to BoNT/A preparation 1, but the development of a neutralizing antibody induced secondary treatment failure in both cases and a totally different response after switch to BoNT/A preparation 2. Conclusions: These five patients indicate that the response of a patient to a BoNT preparation may be unexpected. Therefore, cautious onset of BoNT therapy is recommended as well as consequent dose adjustment later on and even switch to another BoNT/A preparation when a patient has already developed NABs against BoNT/A.

## 1. Introduction

The potent Botulinum Neurotoxins (BoNTs) are metalloproteases that specifically cleave N-ethylmaleimide-sensitive factor attachment protein receptor (SNARE) proteins in synaptic terminals, resulting in a potent inhibition of vesicle fusion and transmitter release [1]. BoNTs are produced by strains of Clostridium botulinum [2]. The natural target of these toxins is represented by the neuromuscular junction, where BoNTs block acetylcholine release. BoNTs also block exocytosis of a variety of glands. Direct delivery to the central nervous system (CNS) [1] and blocking of several transmitters, leads to a near-complete silencing of neural networks.

BoNTs are among the deadliest natural substances known to humans with an estimated lethal dose of 1 ng/kg [3]. High concentrations (>50 ng/person) of some of these neurotoxins cause a life-threatening paralyzing disease called botulism [4]. Toxicology data are available only for a limited number of the many BoNTs so far identified and only for few vertebrate species [5]. However, considering the data limitation, there is still a scattered scheme of responses to BoNTs among different vertebrate species.

A lethal dose, 50% (LD50) is the amount of toxin that can kill 50% of the tested population after administration via different routes and after a specified test duration. The route of application can be intramuscular injection, intraperitoneal injection, inhalation or intravenous injection. In Table 1, LD50 values of tetanus toxin and BoNT/A and BoNT/B for different mammals are presented in relation to the LD50 mouse value. In rats as well as chickens, a mutation at the site of tetanus neurotoxin cleavage of Vesicle-Associated Membrane Protein 1 (VAMP1) leads to a complete resistance to tetanus neurotoxin whereas humans are highly sensitive to tetanus toxin (Table 1; first row) [5]. Additionally, for BoNT/B, rats appear to be highly insensitive, comparably to humans, whereas rabbits seem to be highly sensitive to BoNT/B injections. Against BoNT/A rats are much more sensitive than against tetanus toxin or BoNT/B; however, rabbits are even more sensitive to BoNT/A than rats, mice or humans. These data imply that the species-dependent difference of the sensitivities are genetically determined, and that results on sensitivity of a species to one BoNT cannot and may not be transferred to another species or BoNT.

In clinical practice it is very important to know how safe BoNT/A or BoNT/B therapy is. The LD50 values for A1 progenitor toxin (BOTOX^®^; Allergan Inc., Dublin, Ireland), e.g., in rats and mice are 117 and 93 U/kg, respectively [6]. Thus 1 vial Botox^®^ contains enough toxin to kill two rats (mean body weight (BW) = 400 g) or 35 mice (BW = 30 g). To reach the LD50 of humans (1.7 ng/kg) application of about 2.5 × 70 = 175 vials of Botox (0.73 ng clostridium neurotoxin per 100 U vial [7]) or 17,500 U are necessary. This implies that the clinical application of 200 to 400 U is safe and far from a life-threatening dose.

However, in clinical practice of BoNT therapy, not only knowledge about LD50 is relevant, but also the efficacy and the side effects. Most studies aim at the typical patient who has a clear clinical benefit with a tolerable spectrum of adverse events. The extreme ends of the response spectrum are often neglected. However, knowledge about these extremes of the clinical spectrum of response may be helpful to learn more about the pharmacology of the BoNTs.

The distribution of responses is broad and dependent on the ways of response measurement or efficacy. However, in general, three categories of responders can be distinguished: (1) the bad responder, (2) the usual responder and (3) the super responder. It is highly likely that the three responder subtypes differ genetically, however this has not been analysed in detail so far. We want to focus on the extremes of response since these patients may contribute key aspects for the understanding of BoNT action in clinical practise.

Already the therapeutic decision made at the onset of BoNTA therapy can have relevant implications for the later long-term treatment. In those patients who develop a secondary treatment failure (STF) later on the response to the second injection was reduced in comparison to those patients who did not develop a STF [8]. Therefore BoNT/A therapy should not be started with high initial doses to avoid neutralizing antibody (NAB) induction already during the very first injections. It is not an easy question with which dose BoNT/A treatment should be started. This will be demonstrated by a patient being very sensitive to aboBoNT/A. When BoNT/A therapy is started with a low dose, dose adjustment is very important, otherwise patients will be treated suboptimally [9]. This will be demonstrated by the two following cases. In one patient, incoBoNT/A dose was doubled after the first injection with a standard dose of 200 U incoBoNT/A and did not have any effect. In the other patient, the dose was increased from 800 to 1000 U aboBoNT/A and the patient responded to the surprise of the treating physician after years of BoNT/A treatment.

However, beside optimization of BoNT application, one of the most challenging problems in long-term BoNTA therapy is the avoidance of NAB induction. To place the needle into the most relevant structure for the treatment effect different guidance techniques have been developed. Ultrasound (US), Computertomography (CT), and Electromyography (EMG) guidance are available to administer BoNT exactly. However, induction of NABs can hardly be avoided by special techniques since the immune system cannot be overcome simply by the use of special techniques. The best strategy is to reduce the clostridial protein load per session and to avoid short injection cycles [10,11,12].

In the following, we focus on these three aspects: response at onset, response to dose adjustment, response when a STF has developed. In all three aspects, we concentrate on the extremes of response.

## 2. Patients: The Concept of Treatment Response Variability to BoNT over Time

The relevance of the extreme response behaviour to BoNT will be demonstrated by presenting five different cases:The typical supersensitive patient;The insensitive patient to a standard dose being highly sensitive to a much higher dose;The typical insensitive patient;The patient with complete STF being highly sensitive after switch to incoBoNT/A;The initially highly sensitive patient with CSTF without response to incoBoNT/A.

### 2.1. Response at Onset (General Considerations)

BoNT therapy is usually started with a recommended standard dose. Such a standard dose is suitable for the majority of patients. However, it is too low for an insensitive and too high for a supersensitive patient. This will be explained by two examples:

#### The Typical Supersensitive Patient

This middle-aged female had complaint of neck pain and muscle tension in shoulder and neck areas over two decades. She was seen by orthopaedic surgeons and neurosurgeons without any clear-cut pathological finding. She had a car accident and a whiplash injury in her medical past history. About two years later, the muscle tension in her neck area increased and her head was turned to the right and severely retracted. According to the cap-col concept [13] she suffered from a torticaput to the right, a retrocaput, and a retrocollis. According to the classical classification she had torticollis to the right and a severe retrocollis (comp. Figure 1). The TSUI-score was 12 at that time. Her body weight was more than 100 kg. The treating physician started BoNT/A with 500 U aboBoNT/A (in a dilution of 200 U/mL) but explained to the patient that he suspected that this dose would be too low for a relevant improvement. The course of disease was documented by standardized handy videos or pictures and TSUI scores at every injection visit.

After 1 week the patient became unable to lift the head. This neck weakness persisted for about 3 months and slowly disappeared during the next 2 months. The patient was highly satisfied with the BoNT/A therapy. She was reinjected with 360 U aboBoNT/A after 5 months and again experienced a severe neck weakness which persisted 2 months and then slowly disappeared. She was reinjected again with an even lower dose of 200 MU aboBoNT/A and again experienced a neck weakness. In the meantime, the TSUI score decreased from 12 to 2. When the patient was injected with 100 MU aboBoNT/A (50 U/mL), she also experienced a mild neck weakness. When the patient had been switched to 100 U incoBoNT/A (50 U/mL) no neck weakness occurred and the duration of the effect was shorter than 3 months so that the dose of incoBoNT/A had to be increased to 150 U. After four injections with incoBoNT/A (50 U/mL), she was injected again with 100 MU aboBoNT/A (50 U/mL) and again experienced a mild neck weakness and a much longer duration of effect than after incoBoNT/A. Therefore, she wanted to continue with BoNT/A injections every 4 to 5 months with 100 U aboBoNT/A (Figure 1).

### 2.2. Response during Dose Adjustment (General Considerations)

A variety of patients are transferred to our outpatient clinic because of poor response to the first one to three BoNT injections, apart from the wrong muscle selection by not taking into account the cap-col concept [13], for example, a too low dose is the most frequent cause for a primary non-response [9]. Since high dose per session is a risk factor for NAB induction, we switch these “primary non-responders” to high doses of incoBoNT/A. Most of these patients show a good response when the BoNT/A dose per session exceeds a critical limit. This is demonstrated by the next patient.

#### 2.2.1. The Insensitive Patient to a Standard Dose Being Highly Sensitive to a Much Higher Dose

This 32-year-old male IT specialist did not have any symptoms before the age of 30. In January 2019, he first realized that his head tended to turn to the right. Within 3 months a severe cervical dystonia (CD) developed with a torsion component to the right, a tilt to the left and a retrocomponent. The TSUI score was 8. BoNT/A therapy was initiated at his 31st birthday with a standard dose of 200 U incoBoNT/A (dilution: 50 U/mL). Course of disease was documented by the patient who performed standardized handy photos. For 3 weeks the patient experienced a mild improvement. Thereafter, CD worsened again and the head turned to the right by about 90 degrees 3 months later. The dose was increased to 400 U incoBoNT/A (dilution: 50 U/mL). He responded very well to this increased dose and the TSUI score decreased to 4 three months later. During the next 3 injections with 400 U incoBoNT/A improvement further continued. After five injections and 1 year and 3 months after onset of BoNT therapy, the TSUI score was zero and the patient reported an improvement of 95%.

#### 2.2.2. The Typical Insensitive Patient

This 55-year-old male post officer suffered from psoriatic skin problems from his youth on. After the development of neck pain and muscle tension in neck area that made head-turn to the right, torticollis was diagnosed and BoNT/A therapy was started with 800 MU aboBoNT/A (dilution: 200 U/mL) at the age of 40. Course of disease was documented daily by the patient using a visual analogue scale (VAS: 0–100). The TSUI score was 8. The pain in the neck decreased and the patient had the impression that the force which pulled the head to the right had decreased. During the next years, BoNT/A therapy was continued by application of 800 MU aboBoNT/A, the TSUI score fluctuated between 8 and 10. The patient always reported a consistent reduction of neck pain, but never a clear-cut improvement of head position.

Presence of NABs was analysed six times. The MHDA revealed a threshold near positive value only once. The following confirmation tests were all negative. Around that time, CD progressed and a Meige syndrome and oropharyngeal dystonia developed. The dose was increased to 1000 MU aboBoNT/A (200 U/mL), the patient responded and the TSUI decreased to 6. The patient has developed a neck muscle atrophy on the right side. The patient rates the therapy effect by 30%. However, he wanted to continue BoNT/A therapy.

### 2.3. Response after Development of Neutralizing Antibodies or Secondary Treatment Failure (General Considerations)

It would be the optimal response behaviour if a patient responds almost consistent to BoNT/A therapy over years. When a patient starts to report that the efficacy of the injection therapy declines and also the treating physician observes a worsening then the development of a NAB-induced secondary treatment failure can be suspected [8,14]. The patient with PSTF should be switched to incoBoNT/A as soon as possible to avoid high NAB titers. Despite a high NAB titre, an excellent response may occur. This is demonstrated by the following patient.

#### 2.3.1. The Patient with Complete STF Being Highly Sensitive after Switch to incoBoNT/A

This 53-year-old theologist experienced muscle pain on the right side of neck area for the first time at the end of 2006 when he was 44 years old. In January 2007, a cervical dystonia was diagnosed in another institute. Laboratory tests for Wilson’s disease and a cranial MRI scan were normal. AboBoNT/A therapy was started with 300 MU (dilution: unknown) without any effect. Less than 2 months later, aboBoNT/A was increased to 625 MU which had a good effect but caused dysphagia and neck weakness. During the next injections, the dose was further increased; nevertheless, dysphagia disappeared after the 4th injection. One year and 3 months after successful treatment with 700 to 800 U aboBoNT/A the patient reported the first time that he had neck pain again for at least 6 weeks. Dose was increased to 1000 U aboBoNT/A. Again, he reported a dissatisfying response and dose was decreased to 800 MU. In November 2011, for the first time, a retrocaput component was noticed by the patient and treating physician. The dose was increased up to 1025 U aboBoNT/A. In September 2012, an additional elevation of the left shoulder occurred, but the effect on neck and shoulder muscles was satisfying. During the next 4 years, dose per session and injection scheme was constant. In September 2017, 1325 U aboBoNT/A were injected without satisfying effect. The retrocomponent worsened and dominated the clinical presentation. In March 2018, the patient was injected with 1450 U aboBoNT/A (dilution: unknown) without any effect. In April 2018, about 11 years after the onset of aboBoNT/A therapy, an MHDA test was performed and revealed “a very high MHDA titre” and cessation of BoNT/A was recommended by the laboratory.

In May 2018, the patient was referred to our institution. He presented with complex cervical dystonia with a torsion component to the right (4 points), a tilt to the left (2 points), a retrocomponent (4 points), and a dystonic head tremor (2 points). In June 2018, the patient was injected for the first time with incoBoNT/A 500 U (dilution: 50 U/mL). Course of disease was documented daily by the patient using a VAS (0–100). The tremor disappeared. Three months later, the TSUI score was 10 and BoNT/A was continued with 500 U incoBoNT/A. Although the torsion component improved the patient was injected with 600 U incoBoNT/A (50 U/mL) the next time. Under four further injections with 600 U incoBoNT/A the patient further improved and reached his best improvement in November 2019. The TSUI score was 2.

Thereafter, the course of disease became less satisfying because of the interference with the COVID 19 pandemic induced lockdown of the botulinum toxin clinic. Treatment cycles had to be prolonged. As a consequence, the patient slightly worsened and had a TSUI score of 4 in August 2020.

#### 2.3.2. The Initially Highly Sensitive Patient with CSTF without Response to incoBoNT/A

This 53-year-old female post officer developed the same symptoms as her mother when she was 35 years old. Her mother had already been treated with BoNT/A in our institution for years when her daughter also developed torticollis to the right, a tilt to the right and a severe dystonic head tremor. The TSUI score was 12. She was included in a study on the treatment of de novo-CD patients with 500 U aboBoNT/A (dilution: 200 U/mL). Course of disease was documented by TSUI scores at each injection visit. She responded excellently and the TSUI score dropped down to 0 after 6 injections. During the next years, the low dose was reduced to 360 U aboBoNT/A (200 U/mL) and the TSUI score fluctuated between 0 and 2.

In 2012 head tremor worsened and the TSUI score was 4 and the patient reported that the duration of effect was shorter than before. The next injection was performed with 640 U aboBoNT/A (200 U/mL) and the patient reported a further worsening of head tremor. In January 2013 when the TSUI score reached a value of 6 she was switched to 300 U incoBoNT/A (dilution: 50U/mL) and a blood sample was taken for the MHDA test. The patient did not experience a relevant effect of this first incoBoNT/A injection and wanted to continue BoNT/A with aboBoNT/A again. She was switched back to 800 U aboBoNT/A (200 U/mL). At the next visit, the TSUI score was 2, but the head tremor worsened again and the TSUI score further increased when aboBoNT/A therapy was continued. At the beginning of 2014, the result of the MHDA-test was available and confirmed NAB-induced STF with a high titre. She was switched again to incoBoNT/A (400 U; dilution 50 U/mL). The 400 MU of incoBoNT/A were without relevant clinical effect and BoNT/A was carried on with 950 U aboBoNT/A (100 U/mL). This dose had a transient effect for 2 weeks. The TSUI score further increased and the head started to turn to the left. From the beginning of 2015 onwards, she was permanently treated with incoBoNT/A with 400 U (50 U/mL). In 2016, the TSUI score was 4, but the next injection was without effect so that she was switched to 1000 U aboBoNT/A (100 U/mL) again. In 2017, both the treating physician and patient agreed that inco- and aboBoNT/A have no effect. Under 500 U, incoBoNT/A TSUI score did not worsen. Primidone turned out to be helpful. In May 2020, the TSUI was 10; the patient did not want a deep brain stimulation operation.

## 3. Discussion

### 3.1. Insensitive and Supersensitive Patients at the Onset of Therapy

Dose selection at the onset of therapy is not a trivial problem. Standard doses for the different preparations (e.g., for patients with CD: 500 U for aboBoNT/A (SPC, 8 April 2015); 200 U for onaBoNT/A (SPC; revised July 2020); 200 U for incoBoNT/A; 5000 for rimaBoNT/B (26 February 2014) are recommended on the basis of studies which were designed to yield a satisfying effect in as many patients as possible with tolerable adverse events. In these studies, complex patients were often excluded which may need individual adjustments of injection scheme and dose. For example, in most patients with a pure antecollis or antecaput, severe dysphagia occurs when the deep anterior neck muscles are injected regardless whether EMG or CT guidance is used [15].

Phase III studies usually do not take into account the patient’s history before BoNT/A therapy. It has recently been demonstrated that a low initial severity is often interpreted as a hint for low disease activity with the consequence that low doses were applied. This leads to further worsening and cessation of BoNT therapy. For patient management, it is important to know that a low initial severity of the disease may be a hint for a still progressing disease which has to be treated with a sufficiently high dose [9]. In Figure 2, an example for such a patient is demonstrated. This patient with a rapidly increasing disease at the first visit had only a minor response to 200 incoBoNT/A. Progression could not be stopped by means of this standard dose.

On the other hand, the example in Figure 1 demonstrates that a standard dose may be much too high for a supersensitive patient. The reasons for supersensitivity to BoNT/A are poorly understood. A patient may suffer from undiagnosed myasthenia gravis [16] or Labert-Eaton-syndrome and therefore may be highly sensitive to BoNT. However, the example in Figure 1 was highly sensitive to aboBoNT/A and not to incoBoNT/A, but did not suffer from myasthenia gravis. This patient responded as if 1 aboBoNT/A unit was equipotent to 1 incoBoNT/A, compared to the response of a usual patient, this patient seems to be 10 times more sensitive to aboBoNT/A. We have observed such supersensitivity also in a patient with blepharospasm. However, in the literature, little is documented about such patients.

### 3.2. Supersensitive and Insensitive Patients during the Phase of Dose Adjustment

The example in Figure 1 demonstrates that it may take time to find the optimal dose to keep the patient compliant. Additionally, the example in Figure 3 underlines that it may be important for an optimal outcome to adjust the dose also during long-term treatment.

Most of the diseases being treated with BoNT are not stable over time. Both spasticity, as well as dystonias, may progress and patients experience new symptoms or worsening of already present symptoms [17]. Therefore, an adjustment of dose may be necessary to maintain a stable level of improvement.

The reasons for the progression of dystonias are unclear. It may very well be that blepharospasm, e.g., is the first sign of a Meige syndrome, which fully develops over years after the onset of BoNT therapy of the blepharospasm. It is well-known that, in about one third of the CD patients, new symptoms come up during the course of BoNT/A which had not been present at the onset of BoNT/A therapy [17]. In about 16% of the patients, segmental dystonia develops during the course of treatment [18]. Obviously, the symptomatic treatment of BoNT injections does not stabilize the disease progression in the central nervous system [17].

Furthermore, it may very well be that in addition another neurological disease develops which also leads to worsening of symptoms of pre-existing dystonia or spasticity. We have observed progression of symptoms of CD in patients in whom multiple sclerosis was diagnosed after CD had become manifest. Polyneuropathies may worsen spasticity. Thus, there are various reasons to adjust the dose during long-term treatment.

However, to our experience, the most frequent reason to adjust the BoNT dose during long-term treatment is the development of a partial secondary treatment failure [19].

### 3.3. Sensitive and Insensitive Patients to Switch of the BoNT/A Preparation after Induction of NABs or STF

Clinical efficacy follows a typical course. A few days after injection, the clinical effect of a BoNT/A injection increases rapidly, reaches a maximum after about 3 to 5 weeks, and decreases again. The duration of action depends on disease entities, patients, doses, and the precision of injections. When a further injection is performed before the effect of the previous injection has fully declined, an even better result can be observed than after the previous injection has fully vanished. However, with repeated injections, the risk of antibody (AB) induction will increase [12,20,21].

Antibodies against BoNT can be broadly divided into neutralizing antibodies (NABs), targeting the core neurotoxin, particularly the binding site on the heavy chain, and non-neutralizing antibodies, typically targeting accessory proteins or clinically irrelevant sites on the core neurotoxin and which do not affect clinical efficacy [14,22]. Various laboratory assays have been used to detect antibodies in patients with possible immunoresistance. Bioassays such as the Mouse Protection Assay (MPA) or Mouse Hemidiaphragm Assay (MHDA) utilize animals to identify neutralizing antibodies that impact the clinical efficacy of the toxin [23].

However, so far it is unclear whether bioassays are more sensitive than careful clinical observation. To our experience, it is more likely that subthreshold NABs have already induced PSTF in patients who report a lack of efficacy and show a worsening in objective scales than to assume the occurrence of another unknown reason for the worsening [24].

In patients who reliably worsen despite an increase of dose the BoNT preparation should be switched to a complex protein free BoNT/A preparation [25] well before a complete secondary treatment failure was developed [26]. In most of the partial secondary non-responders, switching to rimaBoNT/B yields only a transient improvement. This has been observed for the use of higher [27] and lower rimaBoNT/B doses [28].

Switching to another BoNT/A preparation is usually not recommended [14]. A successful restart of BoNT/A therapy after cessation of BoNT/A therapy for years has been reported [29]. To our experience, NAB titres may decline and patients may have a good clinical response when they are switched to incoBoNT/A immediately after NABs have been detected [30].

This is demonstrated by the example in Figure 4. Immediately after high titers of NABs against aboBoNT/A had been detected the patient was switch to incoBoNT/A.

On the other hand, the example in Figure 5 demonstrates that not all patients who had responded highly sensitive in the beginning and had developed CSTF also respond well to the switch to incoBoNT/A. This patient had become a secondary complete insensitive patient. Due to the large response spectrum after the switch to incoBoNT/A in patients with STF, dose increase and switch to incoBoNT/A is a “must” and an alternative to deep brain stimulation.

### 3.4. Future Investigations on Insensitive and Supersensitive Patients

We have emphasized the extremes of response in the course of BoNT therapy. Figure 6 gives an overview of the different aspects of the present paper. Thus far, little is known about the reasons why clear differences in the response to BoNTs exist. Differences between different species have a genetic background. We, therefore, think that differences in response within a species are also genetically determined.

This is confirmed by the analysis of different MHC classes and the demonstration that some MHC classes have a different distribution in NAB-positive and NAB-negative patients [31]. To improve the sensitivity and specificity of such tests, we recommend comparing insensitive and supersensitive patients, since these different patient groups (see Figure 6) may reveal clear differences.

We, therefore, recommend sharpening the view for the difference in response behaviour and to define insensitive and supersensitive patients more clearly to get an estimate of how frequent insensitive and supersensitive patients are, and to determine their percentage in a larger population of BoNT-treated patients for further testing of the genetical background.

## 4. Conclusions

The supersensitivity/insensitivity problem (extremes of response to a standard intervention) is present during the entire course of BoNT/A treatment. It becomes obvious at the onset of BoNT/A therapy, it has relevance during the long-term treatment and is present even in the situation when patients have developed NABs against a BoNT preparation. It should be known that extremes of response may occur after the switch to another BoNT preparation.

## Figures and Tables

**Figure 1 toxins-13-00022-f001:**
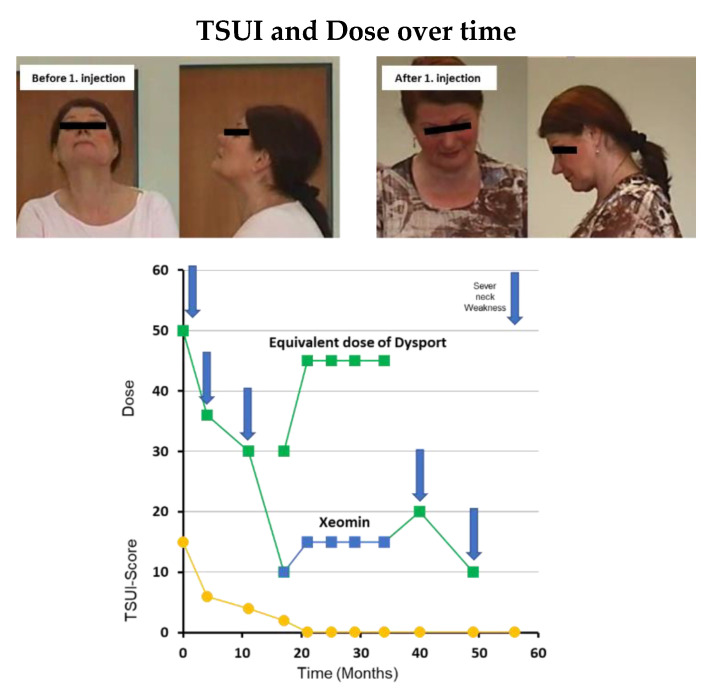
These are photos from the typical supersensitive patient before and after first injection. She had torticollis to the right and a severe retrocollis with TSUI of 12 before and neck weakness and inability of lifting head after first injection. (Circle: TSUI score; square: dose; blue: incoBoNT/A; green: aboBoNT/A).

**Figure 2 toxins-13-00022-f002:**
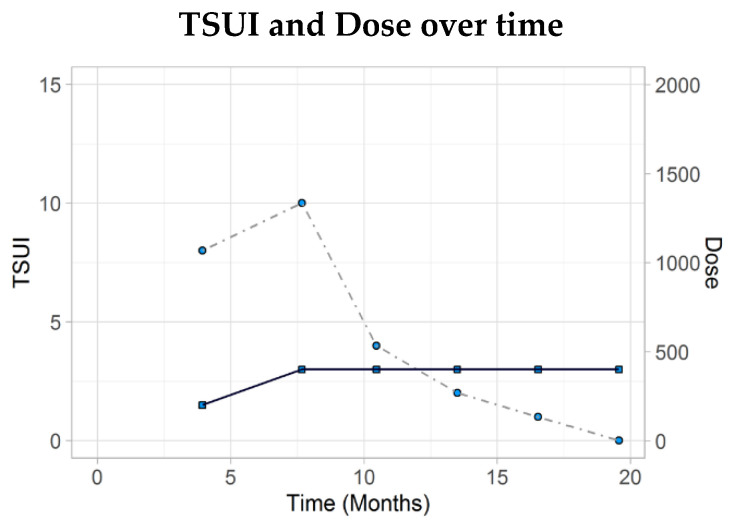
This 32-year-old male did not respond to 200 U incoBoNT/A but revealed a continuous improvement after 400 U incoBoNT/A.

**Figure 3 toxins-13-00022-f003:**
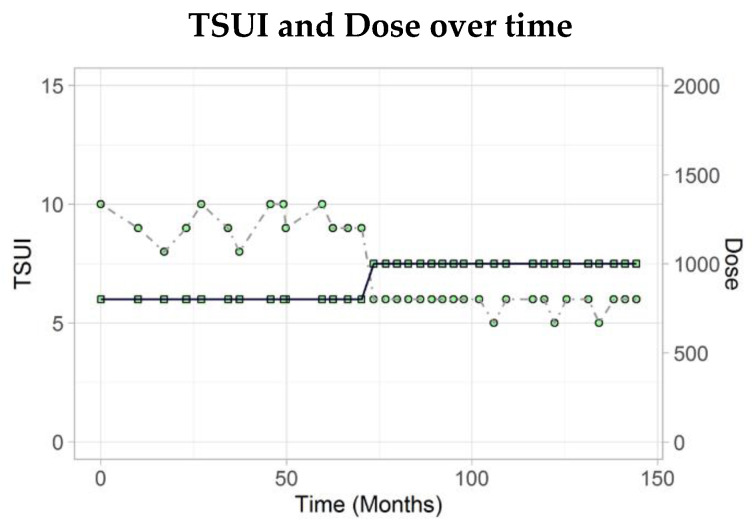
The patient was treated for about 6 years with 800U aboBoNT/A without significant reduction of the TSUI score. After increase of aboBoNT/A to 1000U, the patient responded and the TSUI score decreased to 6.

**Figure 4 toxins-13-00022-f004:**
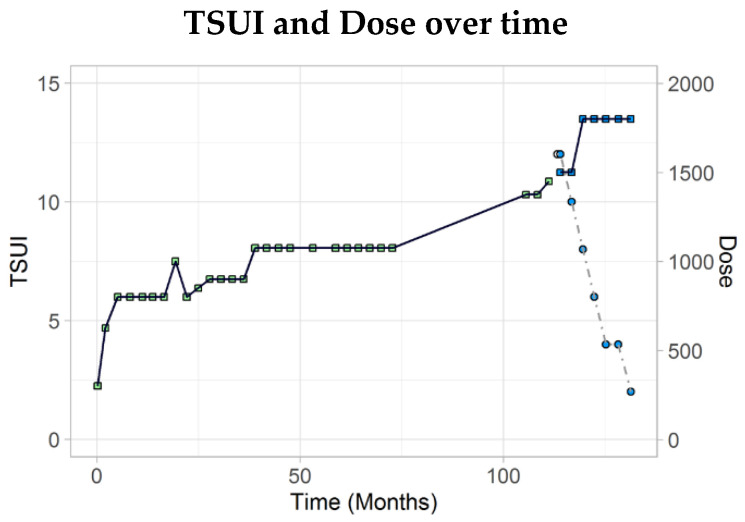
Worsening in a 53-year-old patient who developed a NAB-induced CSTF under aboBoNT/A therapy. The patient responded excellently to high dose of 500 to 600U incoBoNT/A (dose ratio: aboBoNT/A:incoBoNT/A 3:1).

**Figure 5 toxins-13-00022-f005:**
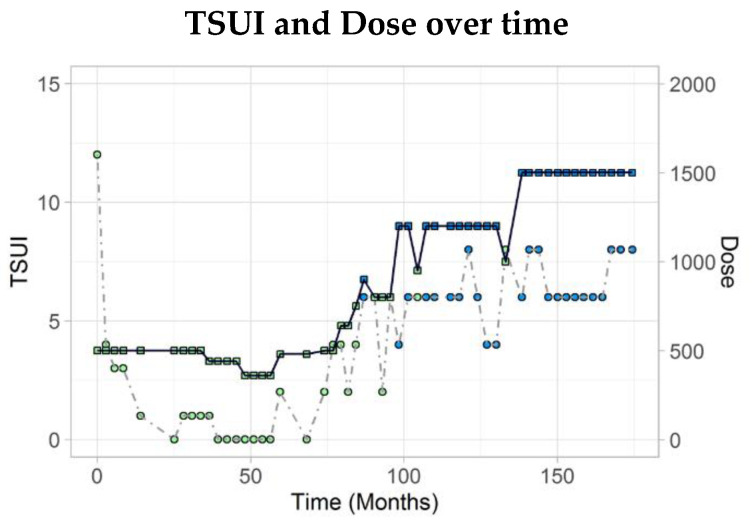
This 53-year-old female responded excellently to aboBoNT/A at onset of BoNT/A therapy. Then she developed a NAB-induced PSTF. Switch to incoBoNT/A was without improvement.

**Figure 6 toxins-13-00022-f006:**
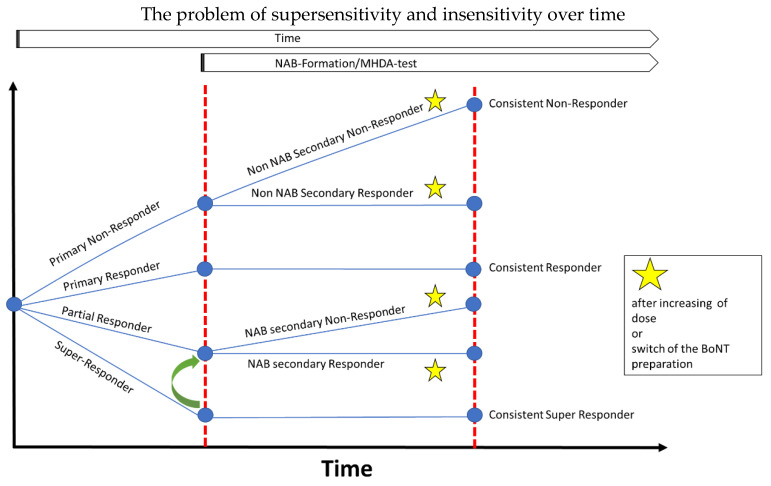
Overview: the supersensitivity/insensitivity problem has to be taken into account at onset of BoNT therapy, later on during dose adjustment and when a secondary treatment failure has developed.

**Table 1 toxins-13-00022-t001:** Minimal mouse lethal doses of tetanus toxin and toxicity of BoNTs in different mammals.

		Mouse	Rat	Rabbit	Monkey	Human
Tetanus Toxin (ng/kg)		0.5 ng/kg (IM)	resistant	3 ng/kg (IM)	0.4 ng/kg (IM)	0.2 ng/kg (IM)
Botulinum Toxin (* expressed as multiple of the mouse LD50/kg)	BoNT/A	1 (IP) * 1 (IM) *	2.5 (IP) *	0.3 (IP) * 0.8 (IM) *	0.78(IM)*	1.3–2.1 ng/kg (IM, IV) 10–13 ng/kg (INH) 1000 ng/kg (oral route)
	BoNT/B	1 (IP) * 1 (IM) * 1 (INH) *	1000 (IP) *	0.1 (IM) *	150-432(INH)*	

Adapted from [5], Tetanus toxin toxicity value for humans is the results of extrapolation from monkey data. IM, intramuscular; IP, intraperitoneal; INH, inhalation; IV, intravenous. This table is a selective part of data shown in review article by Ornella Rossetto and Cesare Montecucco with small modification relevant to this study [5].

## Data Availability

Data available on request due to restrictions eg privacy or ethical. The data presented in this study are available on request from the corresponding author.

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
