# Peer review of "The Extreme Ends of the Treatment Response Spectrum to Botulinum Toxin in Cervical Dystonia"

_toxins, 2020, doi:10.3390/toxins13010022_

Round 1

Reviewer 1 Report

This manuscript is not suitable for consideration in the current form.

The use of comments about "rats" and "rabbits" is completely inappropriate and nonsense.  It trivialises the entire subject of different patient responses to botulinum toxin and is demeaning to the patients described.  It is also confusing to the reader as to whether the authors mean animal responses or human responses!

I recommend that this manuscript be rejected and the authors told to rewrite the content, if they wish it to be taken into further consideration.

Author Response

Thank you so much for your time and comments.

We now avoid the words „rats“ and „rabbits“ throughout the text. Only at two places in the text we mention that in our team language „rats“ and „rabbits“ stand for poor and good responders. 

Reviewer 2 Report

The authors highlight the spectrum of possible responses to botulinum toxin in cervical dystonia and focus on the extremes of this spectrum with clear and well documented examples. The paper is well researched and clearly written. As they mention, the emphasis on most publications is on common, relative simple cases of cervical dystonia but to my knowledge there are no publications concentrating  on the more challenging cases of cervical dystonia that have an unusual response to botulinum toxin and I think this paper will be of help and interest to relatively new and experienced injectors.

Author Response

Thank you so much for your time and comments.

Reviewer 2 has picked-up the reason for presenting this case series quite well: to focus on the extremes of the response spectrum.

Reviewer 3 Report

Manuscript presented by authors is very interesting and innovative; authors deal with the problem of botulinum toxin type A treatment of non-responders in an original way, considering five patients and identifying them with a different pattern (from rat to rabbits). However, some changes need.

Abstract

Minor corrections of the English language

Introduction

Well structured. On line 69 and line 70, STF and NAB are respectively written; authors should first explain in the text what they mean by those acronyms. Lines 80-81 talk about the various injection methods; I would conclude the sentence indicating acronyms which are then found later in the text. Minor corrections of the English language.

Patients

Presentation of five cases is well structured. However, for each case the authors should indicate:

-Dilution of each treatment

-Method of inoculation (ultrasound or electromyography or others) of each treatment

- If other clinical and / or instrumental parameters were evaluated (electromyography, ROM, etc.).

Minor corrections of the English language

Discussion

Well structured. I would add more comments between the study and even more recent literature. Minor corrections of the English language

Author Response

Reviewer 3

Manuscript presented by authors is very interesting and innovative; authors deal with the problem of botulinum toxin type A treatment of non-responders in an original way, considering five patients and identifying them with a different pattern (from rat to rabbits).

However, some changes need.

Indeed, the present paper is an attempt to deal with the problem of extremes of the response spectrum in an innovative way.

We have taken into account all points raised by reviewer 3.

Abstract

Minor corrections of the English language

Has been performed.

Introduction

Well structured.

On line 69 and line 70, STF and NAB are respectively written; authors should first explain in the text what they mean by those acronyms. Lines 80-81 talk about the various injection methods; I would conclude the sentence indicating acronyms which are then found later in the text.

Minor corrections of the English language.

Now STF and NAB are explained before these abbreviations are used. In our version of the manuscript this problem was found in line 169 and 170.

In lines 180 and 181 we now introduce abbreviations (US-, CT-, EMG-) which were used later on.

Has been performed.

Patients

Presentation of five cases is well structured. However, for each case the authors should indicate:

-Dilution of each treatment

-Method of inoculation (ultrasound or electromyography or others) of each treatment

- If other clinical and / or instrumental parameters were evaluated (electromyography, ROM, etc.).

Minor corrections of the English language

We are thankful for these helpful comments:

-Dilution is now mentioned.

-For injection of patients with CD in our ambulance we do not use special guidance techniques.

-Patients are instructed to assess efficacy of treatment by standardized handy photos or videos or by daily scoring using a visual analoque scale (0-100). Treating physicians assess the treatment effect using the TSUI-score at each treatment visit.

This is mentioned in the text now.

Has been performed.

Discussion

Well structured.

I would add more comments between the study and even more recent literature.

Minor corrections of the English language

We have added two more references from 2020, dealing with the frequency of complete

Secondary treatment failure (Walter et al. 2020) and the clinical implications of possible differences in antigenicity of different BoNT/A preparations.

Has been performed. 

Round 2

Reviewer 1 Report

In the revised version, the authors have still continued to use the RAT and RABBIT analogy.  This was highlighted to them in my first review report, but they have ignored my comment.  I fundamentally disagree with this labelling.  It is trivialising the patients and introducing extra complexity to a simple series of case reports.

I recommend that the manuscript be rejected.

I have not reviewed the manuscript as I would do normally, therefore there may be additional comments.

Author Response

Dear Sir/ Madam,

thank you for the invaluable chance to revise the paper. Regarding suggestions, we have omitted all cases indicating either explicitly or implicitly the relation to animal models (in figures and paragraphs), and we also applied minor edits in some paragraphs. We gratefully hope now that the concerns have been addressed appropriately.

should it be further revision and modification, please do not hesitate to contact us.
